# The Role of Hyperarousal and Aberrant Salience in the Acceptance of Anti-COVID-19 Vaccination

**DOI:** 10.3390/medicina59081403

**Published:** 2023-07-31

**Authors:** Fiammetta Iannuzzo, Rosa De Stefano, Maria Catena Silvestri, Clara Lombardo, Maria Rosaria Anna Muscatello, Carmela Mento, Antonio Bruno

**Affiliations:** 1Department of Biomedical and Dental Sciences and Morphofunctional Imaging, University of Messina, Via Consolare Valeria 1, Contesse, 98125 Messina, Italy; fiammetta.iannuzzo@unime.it (F.I.); mariacate@libero.it (M.C.S.); mmuscatello@unime.it (M.R.A.M.); antonio.bruno@unime.it (A.B.); 2Psychiatry Unit, Polyclinic Hospital University of Messina, Via Consolare Valeria 1, Contesse, 98125 Messina, Italy; rosadestefano@gmail.com (R.D.S.); clara.lombardo1988@gmail.com (C.L.)

**Keywords:** hyperarousal, aberrant salience, COVID-19, vaccination

## Abstract

*Background and Objectives*: This present study was aimed at exploring hyperarousal and aberrant salience in a sample of the Italian general population to understand their possible role in the acceptance of anti-COVID-19 vaccination. *Materials and Methods*: Sociodemographic data questions, the “Acceptance of Vaccination” measure, the Hyperarousal Scale (H-Scale), and the Aberrant Salience Inventory (ASI) were sent as an unpaid online survey to the general population (age range 18–80 years) within the Italian territory. *Results*: The enrolled subjects were divided into two subgroups: “Pro-vax” (*n* = 806; 87.4%) and “No-vax” (*n* = 116; 12.6%). Statistical analysis showed significant differences between groups in the “Education Level” (*p* = 0.001) category, higher in the “Pro-vax” group, and in the ASI “Senses Sharpening” (*p* = 0.007), “Heightened Emotionality” (*p* = 0.008), and “Heightened Cognition” (*p* = 0.002) subscales with the “Total Score” (*p* = 0.015), all higher in “No-vax” subjects. Furthermore, a linear regression model evidenced that only ”Education Level” (β = 0.143; *p* < 0.0001) and “Senses Sharpening” (β = −0.150; *p* = 0.006) were, respectively, direct and inverse predictors of “Acceptance of Vaccination”. *Conclusions*: Our results show that several subthreshold conditions, such as somatosensory amplification, anxiety traits, and panic experiences, should be taken into account by authoritative sources involved in health education, communication, and policy to alleviate public concerns about vaccine safety, for the present and also future pandemics, and to provide more inclusive, informed, and accurate public health preventive and treatment programs.

## 1. Introduction

In 2019, the WHO named coronavirus disease 2019 (COVID-19), a severe acute respiratory syndrome caused by the coronavirus 2 (SARS-CoV-2), identified by the Coronavirus Study Group of the International Committee on Taxonomy of Viruses [1]. After causing the death of thousands of people in the world, COVID-19 was declared a pandemic. Since then, many efforts have been directed toward developing vaccines against COVID-19 to avert the pandemic; currently, different vaccines are available to prevent COVID-19 infection, and some of them have reported efficacy according to the scientific literature or reports submitted to regulatory authorities [2]. However, the availability of effective and safe vaccines is not sufficient, since vaccines have to be extensively accepted to confer population benefits. Successful immunization programs rely on high rates of population coverage, which implies public acceptance of COVID-19 vaccination, a variable that is influenced by trust in the effectiveness and safety of vaccines, in the public health system, and in the policymakers who contribute to the development and introduction of new vaccines [3]. Thus, whereas acceptance of vaccination results from a complex decision-making process that can be potentially influenced by a wide range of individual, sociocultural, environmental, and contextual factors, refusal of vaccination does not have a specific definition and, according to the literature, may be part of a continuum known as “vaccine hesitancy”, defined as “delay in acceptance or refusal of vaccination despite availability of vaccination services” [4]. Vaccine hesitancy is a complex, context-specific construct characterized by different degrees of indecision and encompassing a wide range of factors, such as lack of trust, convenience, complacency, and socio-economic and cultural factors [5]. Vaccine hesitancy seems to vary over time and across countries, and for this reason, the evaluation of inequalities in immunization campaigns could represent a cornerstone in the development of health programs and policies [6]. Individuals who show vaccine hesitancy are a heterogeneous group that includes those who refuse all vaccinations in general or specific vaccines and those who may accept available vaccines yet still remain concerned about them. It has been shown that hesitancy towards COVID-19 vaccines seems to be influenced by demographic (i.e., education, income, ethnicity), environmental (i.e., policies, media), and vaccine-specific factors (i.e., vaccine efficacy, safety, and speed of vaccine development) [7]. At an individual level, the potential roles of working status, personal beliefs, religiosity, political inclinations, gender, educational parameters, and age have been evaluated: female gender, unemployed people, religiosity, lower education, lower age, and non-healthcare working people are associated with a reduced willingness to be vaccinated [8]. The role of political ideology has been shown to be insignificant, although conservatism, rather than being a predictor of vaccine acceptance at the individual level, resulted in being associated with vaccine hesitancy on the group level [9].

Structured religious beliefs have been differently associated with vaccine hesitancy, since several of them were related to vaccine skepticism, with possible negative repercussions on global efforts to contain the COVID-19 pandemic [9]. Finally, economic difficulties, unemployment, and low levels of education were connected with a lower acceptance [8].

Furthermore, in the general population, several psychological constructs, such as personality traits or individual differences in cognitive and affective styles and in disposition toward interpersonal and institutional trust, seem to be related to vaccine hesitancy [10].

Studies on special populations have shown that, despite expectations, high rates of adherence to the COVID-19 vaccination were observed among people affected by mental disorders, with associated psychological features substantially comparable with those found in the general population [11]; however, the identification of the psychological characteristics of people with vaccine hesitancy requires extensive and focused studies with standardized instruments. Fear/anxiety seems to be a core feature of vaccine hesitancy; specifically, the dimension “fear of the unknown” might convert caution into idiosyncratic beliefs in a large percentage of the general population [12,13], whereas functional fear might be related to increased compliance to vaccination [14]. Thus, fear is a complex construct whose components should be further deconstructed in more basic dimensions. One of the psychophysiological measures for fear/anxiety is the construct of hyperarousal, deriving from the deregulation of the autonomic nervous system (ANS) and resulting in sympathetic hyper-reactivity (excess of reaction to stimuli). It is well known that greater psychophysiological reactivity characterizes patients with fear/anxiety (i.e., generalized anxiety disorder, panic disorder, social anxiety) or stress-related disorders (i.e., post-traumatic stress disorder). Nevertheless, hyperarousal should be seen as a dimensional trait in a continuum within the internalizing spectrum among people with the undiagnosed psychiatric condition [15]. Furthermore, fear and anxiety negatively affect cognition and have a role in contributing to aberrant salience, described as the attribution of incorrect and personal significance to neutral stimuli [16]. Based on this background from the literature, the current study was designed to investigate the role of hyperarousal and aberrant salience in a sample of the Italian general population with the aim of exploring possible relationships between these dimensions and vaccine acceptance and/or refusal, the two extremes of the vaccine hesitancy continuum.

## 2. Materials and Methods

### 2.1. Sample

Subjects aged between 18 and 80 years, recruited from among the Italian general population, were included in this study. Exclusion criteria comprised the presence, investigated with specific dichotomous questions (“Do you suffer from mental illnesses?”; “Do you suffer from any medical condition that prevents the anti-COVID vaccination?”), of major psychiatric disorders (schizophrenia spectrum disorders, mood and bipolar disorders, obsessive–compulsive disorders, anxiety, trauma and stress-related disorders, and neurocognitive disorders), organic diseases that may have prevented anti-COVID vaccination, and vaccine indecision. Vaccine indecision was explored by the specific question “Have you had the anti-COVID vaccination?”, and the answer “Yes, but after getting infected or after a family member/friend got infected” was considered an exclusion factor.

### 2.2. Methods

Sociodemographic data questions (gender, age, education level), the “Acceptance of Vaccination” measure, and the chosen psychodiagnostic instruments were included in an online panel for data collection (i.e., Google Forms^®^) then diffused in the Italian territory, along with a call to participate in the study, presented as an unpaid, voluntary, and anonymous online survey released through web advertising, social networks (i.e., Instagram^®^, Facebook^®^), professional and institutional mailing lists (i.e., Linked-In^®^, university webmail), and messaging services (i.e., WhatsApp^®^ and Telegram^®^). The research method avoided missing data, since the online module did not allow a participant to proceed if one response was left unanswered. All participants were provided with written explanations of research aims (“an investigation on potential correlations among certain personality features, e.g., our attitude and our experiences, some anxiety components, and the choice whether or not to carry out the anti-COVID vaccination”) and information on the confidentiality of the procedure, which guaranteed total anonymity (“All information collected will be processed in compliance with current legislation on the protection of personal data (Legislative Decree 1967/2003; EU regulation 2016/679) and will subsequently be aggregated and processed anonymously”); the submission of the complete survey was considered informed voluntary consent. Since the study was conducted with healthy subjects, with data properly anonymized, and informed consent obtained at the time of the original data collection, ethical approval was not required. The survey included all types of available COVID-19 vaccines in Italy (Moderna^®^, Cambridge, UK; Pfizer/BioNTech^®^, New York, NY, USA; AstraZeneca^®^, Cambridge, UK; and Johnson & Johnson^®^, New Brunswick, NJ, USA). Data were collected between February and June 2022.

### 2.3. Measures

The following psychometric measures were used:–The Italian version of the Hyperarousal Scale (H-Scale) [17]: a 26-item self-reported inventory that assesses hyperarousal behavioral traits on a 4-point Likert-type scale (0  =  Not At All; 1  =  A little; 2  =  Quite a bit; 3  =  Extremely). The scale produces a total summation score (HSUM) with scores for ‘Introspectiveness’ (6 items), ‘Reactivity’ (3 items), and ‘Extreme responses,’ referring to the total number of items checked as ‘extremely.’ Higher scores (max. 78) are representative of higher levels of hyperarousal. The Italian version of the H-Scale obtained an alpha value of 0.82, showing good psychometric properties.–The Italian version of the Aberrant Salience Inventory (ASI), a 29-item questionnaire with a dichotomous scale (Yes = 1, No = 0), was used to assess aberrant salience. Scores are assigned using the sum of the ‘yes’ replies. Higher scores indicate stronger aberrant salience. The ASI is composed of five factors: feelings of increased significance, sense sharpening, impending understanding, heightened emotionality, and heightened cognition. The Italian version of the ASI shows good psychometric properties, high internal consistency, and test–retest reliability, as demonstrated by Cronbach’s alpha = 0.89 [18].–The “Acceptance of Vaccination” was assessed with a specific question (“How much are you in favor of the anti-COVID vaccine?”) on a 4-point Likert-type scale (0  =  Not At All; 1  =  A little; 2  =  Quite a bit; 3  =  Extremely).

### 2.4. Statistical Analysis

Continuous data were expressed as mean ± standard deviation, and the differences between the groups were assessed using Student’s *t*-test for independent samples.

Further, all the variables that reached statistical significance were analysed in a linear regression model, where “Acceptance of Vaccination” was taken as a dependent variable, and the demographic and/or psychometric data were entered into the equation as independent variables, to investigate which demographic and/or personality features could play the role of a specific predictor towards the different aspects of the vaccine choice.

To lessen the risk of Type 1 errors, a Bonferroni correction was applied (0.05/5), and a *p*-value ≤ 0.01 was considered statistically significant. The statistical analysis was performed with the Statistical Package for the Social Sciences (SPSS) 25.0 software (SPSS Inc., Chicago, IL, USA).

## 3. Results

A total of 969 subjects submitted a complete survey; 47 surveys have been removed as subjects met the psychiatric and/or medical exclusion criteria (*n* = 24) or showed vaccine indecision (*n* = 23). The final sample consisted of 922 subjects (females *n*/% = 627/68; males *n*/% = 295/32), characterized by a mean age of 40.5 ± 13.3 years (age range 18–80 years) and by an education level of 15.5 ± 2.3 years. According to the answer provided to the specific question “How much are you in favour of the anti-COVID vaccine?” (“No-vax” = score 0/1 vs. “Pro-vax” = score 2/3), the enrolled subjects were divided into two subgroups: (1) “Pro-vax” group (*n* = 806; 87.4%) and (2) “No-vax” group (*n* = 116; 12.6%).

The demographic features and psychometric scores of both subgroups are summarised in Table 1: the Student’s *t*-test showed significant differences between groups in the “Education Level” (Mean age ± S.D. Pro-vax vs. No-vax: 15.65 ± 2.18 vs. 14.87 ± 2.62; *p* = 0.001), higher in the “Pro-vax” group, and in the ASI “Senses Sharpening” (Mean age ± S.D. Pro-vax vs. No-vax: 1.87 ± 1.23 vs. 2.21 ± 1.39; *p* = 0.007), “Heightened Emotionality” (Mean age ± S.D. Pro-vax vs. No-vax: 2.54 ± 1.67 vs. 2.98 ± 1.84; *p* = 0.008), and “Heightened Cognition” (Mean age ± S.D. Pro-vax vs. No-vax: 1.82 ± 1.68 vs. 2.35 ± 1.88; *p* = 0.002) subscales and the “Total Score” (Mean age ± S.D. Pro-vax vs. No-vax: 13.19 ± 6.95 vs. 14.90 ± 7.71; *p* = 0.015), all higher in “No-vax” subjects; the differences in the other variables did not reach a statistically significant level.

All the variables that reached statistical significance (“Education Level”, “Senses Sharpening”, “Heightened Emotionality”, “Heightened Cognition”, and “ASI Total score”—as independent variables) were analysed in a linear regression model in order to evaluate the possible role as predictors towards the “Acceptance of Vaccination” (as dependent variable) (Table 2): as a block, the predictors accounted for 4.8% of the total variance (R = 0.219; F = 9.224; df = 5; *p* < 0.0001). Linear regression analysis indicated that only “Education Level” (β = 0.143; t = 4.294; *p* < 0.0001) and “Senses Sharpening” (β = −0.150; t = −2.728; *p* = 0.006) were, respectively, direct and inverse predictors of “Acceptance of Vaccination”, while the other variables did not make a significant additional contribution to the prediction of the vaccine choice.

## 4. Discussion

The COVID-19 pandemic continues to impose enormous burdens of morbidity and mortality and severely disrupt societies and economies around the world. Overcoming the pandemic is requiring, and will continue to require, effective strategies to increase vaccine confidence and acceptance, which are nuanced with individual differences in personality and cognitive and affective features [19].

The COVID-19 experience led to an unprecedented impact on societies and individuals around the world; for this reason, the biopsychosocial (BPS) model of medicine seems one of the more suitable approaches for dealing with the implications of the pandemic [20]. Within this framework, effective immunization policies should focus not only on the physical health risks of COVID-19 infections but also on the multiple psychological, socio-economic, and individual (beliefs, attitudes) variables affected by the pandemic.

This present study was aimed at exploring hyperarousal and aberrant salience in a sample of the Italian general population to understand a possible link between these constructs and vaccine acceptance or refusal. In our sample, the percentages of “Pro-vax” (87.4%) and “No-vax” subjects (12.6%) were congruent with previous data from the general population showing 77.6% of COVID-19 vaccine acceptation [21] and 15% of COVID-19 vaccine refusal [22].

According to studies showing that fear/anxiety was a core focus of vaccine refusal, we hypothesized that high levels of hyperarousal and aberrant salience, as main components of the fear/anxiety dimensions, may predict vaccine refusal [13].

Our results partially met expectations, since only “Education Level” and “Senses Sharpening”, a component of aberrant salience, resulted as direct and inverse predictors, respectively, of “Acceptance of Vaccination”, whereas the other investigated variables did not provide a significant additional contribution to the prediction of vaccine choice. Indeed, the higher “Education Level” (*p* = 0.001) found in the “Pro-vax” subgroup is congruent with data from previous studies showing that people less favourable to vaccination had lower educational qualifications [8]. Contrary to our hypothesis, hyperarousal levels were not related to the vaccine choice, whereas the role of “Senses Sharpening”, a dimension of aberrant salience, as an inverse predictor of “Acceptance of Vaccination”, deserves further discussion.

The Aberrant Salience Inventory—ASI measures psychotic-like experiences such as magical ideation, and the condition is associated with dissociation and absorption [23]. However, the inventory is composed of several related factors implied in the process of aberrant salience, the abnormally heightened response to neutral or irrelevant stimuli [17]. “Senses Sharpening”, the second factor of the inventory, refers to anomalies of perceptions and subjective feelings of heightened acuteness of the senses [23]. It has been related to schizophrenia’s prodromal symptoms, and it evokes the sensory gating, the ability of the central nervous system to inhibit responses to irrelevant sensory stimuli by the adaptive modulation of its sensitivity to sensory input [24]. However, validation studies of the ASI showed that the factor “Sharpening of Senses” was able to discriminate patients from the general population but not patients with psychosis; it derives that some aspects of aberrant salience may not be psychosis-specific [25]. Abnormal sensory processing patterns have been linked with anxiety in both clinical samples and healthy individuals [26]. The result that the aberrant salience of irrelevant sensory stimuli (“Senses Sharpening”) was an inverse predictor of “Acceptance of Vaccination” is readable if we consider that this dimension indicates a somatosensory amplification, a common factor in conditions related to somatic anxiety, panic, illness anxiety, and fear of bodily sensations [27]. Somatosensory amplification is the tendency to feel a somatic sensation as intense and disturbing. It is associated with several medical conditions characterized by somatic symptoms that are disproportionate to possible organ pathology [28]. In addition, the construct involves an exaggerated negative orientation toward potentially noxious stimuli [29] and this tendency could explain its implication in vaccine refusal.

According to a recent study about cognition, anxiety, and the willingness toward COVID-19 vaccination in tutors of children, people who had not been vaccinated were more likely to have psychological anxiety than those who had been vaccinated, thus demonstrating that anxiety levels influenced vaccine acceptance, although the anxiety scores of all respondents did not exceed the clinical threshold [30]. Generally, anxiety and worries related to vaccination involve many kinds of vaccines: fear of side effects and doubts about vaccine efficacy were the most frequent reasons for refusing influenza vaccines [31]. On the other hand, it has been shown that anxiety and health-related fears were associated with higher vaccine acceptance [32]; these data highlight the need to overcome the broader construct of unspecific anxiety symptoms and to accurately discriminate between different dimensions of fear/anxiety in relation to vaccine acceptance.

This present study has several limitations making the obtained results difficult to generalize. The research sample consisted of Italian subjects. Italy, among the worst-affected countries in the western world by the COVID-19 pandemic, has undergone heavy public health restrictive measures, such as prolonged and repeated lockdowns, significant limitations to travel with closure of regional borders, hygiene measures and social distancing, school closure, and protracted quarantine; nationwide measures to counter the risks of the COVID-19 pandemic have been less restrictive in other countries. The sampling technique (voluntary online survey) may have acted as a spontaneous self-selection bias, since respondents were presumably younger and more skilled with technology when compared with the general population. Moreover, the cross-sectional nature of the study makes it challenging to establish causal relationships from the observed associations. In addition, the “no-vax” group’s size (*n* = 116), although congruent with prevalence rates from the general population, is significantly smaller than the “pro-vax” group, and this may lead to issues with statistical power. Finally, the use of self-assessment measures cannot exclude the possibility of defensive response styles and low self-awareness; thus, it may have influenced the reliability of the responses.

## 5. Conclusions

Nevertheless, results from this study highlight the importance of adding new insight in order to better understand people’s refusal of vaccination, a fundamental measure for the containment of the COVID-19 pandemic. Beyond preventive protective measures and therapeutic approaches, vaccination is the key instrument for containing the pandemic spread of COVID-19; nevertheless, beyond the development and provision of available vaccines, overall COVID-19 vaccination coverage in the general population still remains at suboptimal levels and, despite a general reduction of restrictive measures, the pandemic seems to be far from over [33]. The identification of effective strategies aimed at improving vaccination coverage in the general population requires a comprehensive and deeper understanding of the psychological, cognitive, and emotional variables that contribute to vaccination hesitancy and/or refusal.

Within this context, a general tendency to overlook biopsychosocial interventions in formulating public health policy programs has been reported [34]. Nonetheless, the evaluation of psychological constructs (hyperarousal, aberrant salience), along with other individual and sociocultural features taking into account the biopsychosocial model, could improve educational and advisory healthcare plans, health policy outcomes, and economic burden.

Our results show that several subthreshold conditions, such as somatosensory amplification, anxiety traits, and probably panic experiences, should be taken into account by authoritative sources involved in health education, communication, and policy to alleviate public concerns about vaccine safety, for the present and also future pandemics. Further research should address cognitive beliefs, worries, negative emotions, fear, and anxiety in longitudinal, randomized–controlled studies to provide more inclusive, informed, and accurate public health preventive and treatment programs.

## Figures and Tables

**Table 1 medicina-59-01403-t001:** Descriptive statistics and differences between groups.

	Pro-vax (*n* = 806)		No-vax (*n* = 116)		Student’s *t*-Test	
	Mean	S.D.	Mean	S.D.	T	*p*
Age	40.54	19.52	43.99	11.47	1.859	0.063
Education Level (years)	15.65	2.18	14.87	2.62	−3.493	0.001
H-Scale						
Introspectiveness	9.98	2.68	10.17	2.91	0.711	0.478
Reactivity	3.41	1.73	3.62	1.84	1.205	0.229
Extreme	8.55	9.14	9.47	9.08	1.006	0.315
H-Sum	36.89	8.29	37.22	8.32	0.411	0.681
ASI						
Increased Significance	4.22	1.97	4.42	2.11	1.001	0.317
Senses Sharpening	1.87	1.23	2.21	1.39	2.721	0.007
Impending Understanding	2.61	1.61	2.78	1.57	1.038	0.299
Heightened Emotionality	2.54	1.67	2.98	1.84	2.642	0.008
Heightened Cognition	1.82	1.68	2.35	1.88	3.151	0.002
Total Score	13.19	6.95	14.90	7.71	2.440	0.015

**Table 2 medicina-59-01403-t002:** Linear regression analysis.

		Unstandardized Coefficients		Standardized Coefficients		
Dependent Variable	Predictors	B	S.E.	Beta	t	*p*
**Acceptance of Vaccination ^a^ (Model 1)**	(Constant)	1.719	0.198		8.704	<0.0001
	Education Level	0.049	0.011	0.143	4.294	<0.0001
	Senses Sharpening	−0.093	0.034	−0.150	−2.728	0.006
	Heightened Emotionality	−0.049	0.029	−0.107	−1.668	0.096
	Heightened Cognition	−0.056	0.027	−0.124	−2.090	0.037
	ASI Total Score	0.023	0.012	0.207	1.937	0.053

^a^ R = 0.219; F = 9.224; *p* < 0.0001.

## Data Availability

The data will be available upon request to the principal investigator.

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
