# Peer review of "The Role of Hyperarousal and Aberrant Salience in the Acceptance of Anti-COVID-19 Vaccination"

_medicina, 2023, doi:10.3390/medicina59081403_

Round 1
Reviewer 1 Report
Hello Dears;
Thank you for your good research in the field of vaccination, which is a major challenge in the social and health field.
Comments:
1-Considering the impact of social, economic, religious and cultural factors on vaccine acceptance, please make the article more comprehensive and complete.
2-The type of injected vaccines and level of trust of the people in the target community should be determined in detail.
3-It should be determined whether people with vaccine fear, have other underlying anxiety disorders.
4-in the discussion section, it should be discussed with a comprehensive view of Bio-Psycho-socio- spiritual aspect.
Author Response
To the Editor
Medicina July, 20th, 2023
Ref.: Manuscript ID: medicina-2505473
Title: The role of hyperarousal and aberrant salience in the acceptance of anti-COVID-19 vaccination.
Dear Editor,
Thank you for the opportunity to revise our above-referenced paper. The following letter gives our responses to the Reviewer’s comments. In this revised version of the manuscript, we took into consideration all comments and revised the manuscript accordingly.
Reviewers’ comments:
We wish to thank the Reviewers for reading our manuscript and reviewing it, which has helped us improve it. We next detail our responses to each reviewer’s concerns and comments.
# Reviewer 1
1-“Considering the impact of social, economic, religious and cultural factors on vaccine acceptance, please make the article more comprehensive and complete”.
According to the Reviewer’s suggestion, we have included in the Introduction section a brief overview of the most important social, economic, religious, and cultural factors on vaccine acceptance/refusal.
2-“The type of injected vaccines and level of trust of the people in the target community should be determined in detail.”
The types of Covid-19 vaccines available on the whole Italian territory in the period of the study have been detailed.
3-“It should be determined whether people with vaccine fear, have other underlying anxiety disorders.”
Within the Method section, exclusion criteria have been better specified. The presence of major psychiatric disorders, including anxiety disorders, was investigated with specific questions (page 3, lines 108-112).
4- “In the discussion section, it should be discussed with a comprehensive view of Bio-Psycho-socio- spiritual aspect.”
According to the Reviewer’s suggestion, we have added in the Discussion and in the Conclusion sections a reflection on the appropriate use of the biopsychosocial model as a strategy intervention for the Covid-19 immunization policy.
# Reviewer 2
“The manuscript under review explores the relationship between hyperarousal, aberrant salience, and acceptance of COVID-19 vaccination in the Italian general population. The author(s) demonstrate considerable skill in study design, statistical analysis, and reporting. However, several aspects can be improved for a more comprehensive understanding of the subject.
Strengths:
- The study addresses a topical and pertinent issue, investigating the psychological parameters influencing the acceptance of COVID-19 vaccination. This line of research can help devise more effective vaccination campaigns and address vaccination hesitancy, a pressing problem in public health.
- The investigation of hyperarousal and aberrant salience as potential factors influencing vaccination acceptance adds a novel dimension to the understanding of vaccine hesitancy, expanding upon the traditional sociodemographic factors.
- The methodology appears robust, using reliable scales (H-Scale, ASI) to measure hyperarousal and aberrant salience. The statistical analysis seems well-executed, revealing significant differences between the "Pro-vax" and "No-vax" groups regarding education level and specific ASI subscales. The use of linear regression to identify predictors of vaccination acceptance is appropriate.”
We thank the Reviewer for the positive feedback.
“Drawbacks:
- While the number of participants in the "Pro-vax" group seems satisfactory, the "No-vax" group's size (n=116) is significantly smaller, which may lead to issues with statistical power and the generalizability of the findings.”
We are aware that the "no-vax" group's size (n=116) may lead to issues with statistical power; nevertheless, the size is proportionally congruent with prevalence rates from the general population, and even the increase of the total sample should not modify the numerical group differences. However, this has been stated as a limitation.
- “The paper does not detail how the survey participants were selected. It remains unclear if the sample is representative of the Italian general population, considering age, gender, socioeconomic status, urban/rural distribution, etc.”
We agree with the Reviewer about this issue. Really, although the chosen age criterion was very wide and inclusive for being representative of the Italian general population (18-80 years), the sampling technique may have acted as a spontaneous self-selection bias, since respondents were presumably younger and skilled with technology. This has been listed within the Limitation section.
“The manuscript reports several statistical comparisons (education level and multiple ASI subscales). Without corrections for multiple comparisons, there may be an increased risk of Type I error.”
A Bonferroni correction had been applied (p=0.05/5=≤ .01) in the original version of the manuscript to lessen the risk of Type 1 errors.
- “There is no discussion of missing data. Were there incomplete responses? If so, how were they handled during the analysis?”
Since the online module did not allow respondents to proceed if somewhen one question was left unanswered, there are no missing data and/or incomplete protocols. This has been better described in the Method section.
- “Given the study's cross-sectional nature, it is challenging to establish causal relationships from the observed associations. For instance, it cannot definitively be stated that a lower education level or higher sense of sharpening leads to vaccination hesitancy.”
We agree with the Reviewer that the cross-sectional nature of the study makes challenging to establish causal relationships from the observed associations, and this represents an inner limitation of cross-sectional, punctual studies.
- “Supplementary Materials, Author Contributions, Funding, Institutional Review Board Statement, Informed Consent Statement, Data Availability Statement, Acknowledgments are missed.”
- The required information has been provided, as suggested.
Recommendations:
- “To improve the power and reliability of the study, consider increasing the sample size, particularly for the "No-vax" group.”
The issue has been faced in Drawbacks’ point 1.
- A clear description of the sampling methodology would be beneficial to judge the sample's representativeness. If the sample is not representative, conclusions could be skewed.
See Drawbacks’ point 2.
- “The authors should consider adjusting for multiple comparisons using methods such as Bonferroni correction or false discovery rate to ensure the robustness of the findings.”
See Drawbacks’ point 3.
- “The handling of missing data should be discussed. If missing data are present, Multiple imputations or other similar methods could be considered.”
See Drawbacks’ point 4.
- To better establish a causal relationship between the identified predictors and vaccination acceptance, longitudinal or experimental studies should be considered.
We agree with the Reviewer that cross-sectional are limited in that they do not permit to infer predictions from the obtained data. A wider approach to this topic should consider longitudinal studies, as we had proposed within the Conclusions section.
- “Other sociodemographic variables, such as religious beliefs, political affiliation, and rural/urban residence, can influence vaccine acceptance. Including these variables could provide a more comprehensive analysis.”
We have added in the Introduction section a brief overview of the most important social, economic, religious, and cultural factors that can affect vaccine acceptance/refusal.
“In conclusion, this study contributes to understanding the psychological correlates of COVID-19 vaccine acceptance. However, addressing the abovementioned issues would significantly strengthen the paper and its potential impact on public health.”
Editor’s comments
- “We notice that the repetition rate is higher than our journal requirements. Some high repetition parts are highlighted in the iThenticate report attached. Please revise your manuscript according to the report and make sure that there is no large part repetition with other published papers.”
The iThenticate report has been accurately checked. We found a partial overlap with the methodology of a previously published paper of ours since the recruitment method was similar. However, we have significantly reduced the overlapping parts and the manuscript has been thoroughly revised and checked for repetitions.
- “To facilitate transparent and open science, we encourage authors to publish their results and experimental methodology in as much detail as possible so that results can be reproduced. We noticed that the main text of your manuscript is quite brief which may mean that the materials and methods, research background, future research directions, or possible applications of the research are not described in enough detail.”
We tried to be more detailed in the Methods section. We also added more speculation on the research background, future research directions, and the possible implications/applications of the obtained results.
Finally, we wish to thank the Editor and the Reviewers for their attention and comments on our manuscript.
Again, we appreciate all your insightful comments, and we tried to be responsive to them.
We look forward to hearing from you at your earliest convenience.
Best regards.
Carmela Mento
Reviewer 2 Report
The manuscript under review explores the relationship between hyperarousal, aberrant salience, and acceptance of COVID-19 vaccination in the Italian general population. The author(s) demonstrate considerable skill in study design, statistical analysis, and reporting. However, several aspects can be improved for a more comprehensive understanding of the subject.
Strengths:
1. The study addresses a topical and pertinent issue, investigating the psychological parameters influencing the acceptance of COVID-19 vaccination. This line of research can help devise more effective vaccination campaigns and address vaccination hesitancy, a pressing problem in public health.
2. The investigation of hyperarousal and aberrant salience as potential factors influencing vaccination acceptance adds a novel dimension to the understanding of vaccine hesitancy, expanding upon the traditional sociodemographic factors.
3. The methodology appears robust, using reliable scales (H-Scale, ASI) to measure hyperarousal and aberrant salience. The statistical analysis seems well-executed, revealing significant differences between the "Pro-vax" and "No-vax" groups regarding education level and specific ASI subscales. The use of linear regression to identify predictors of vaccination acceptance is appropriate.
Drawbacks:
1. While the number of participants in the "Pro-vax" group seems satisfactory, the "No-vax" group's size (n=116) is significantly smaller, which may lead to issues with statistical power and the generalizability of the findings.
2. The paper does not detail how the survey participants were selected. It remains unclear if the sample is representative of the Italian general population, considering age, gender, socioeconomic status, urban/rural distribution, etc.
3. The manuscript reports several statistical comparisons (education level and multiple ASI subscales). Without corrections for multiple comparisons, there may be an increased risk of Type I error.
4. There is no discussion of missing data. Were there incomplete responses? If so, how were they handled during the analysis?
5. Given the study's cross-sectional nature, it is challenging to establish causal relationships from the observed associations. For instance, it cannot definitively be stated that a lower education level or higher sense of sharpening leads to vaccination hesitancy.
6. Supplementary Materials, Author Contributions, Funding, Institutional Review Board Statement, Informed Consent Statement, Data Availability Statement, Acknowledgments are missed.
Recommendations:
1. To improve the power and reliability of the study, consider increasing the sample size, particularly for the "No-vax" group.
2. A clear description of the sampling methodology would be beneficial to judge the sample's representativeness. If the sample is not representative, conclusions could be skewed.
3. The authors should consider adjusting for multiple comparisons using methods such as Bonferroni correction or false discovery rate to ensure the robustness of the findings.
4. The handling of missing data should be discussed. If missing data are present, Multiple imputations or other similar methods could be considered.
5. To better establish a causal relationship between the identified predictors and vaccination acceptance, longitudinal or experimental studies should be considered.
6. Other sociodemographic variables, such as religious beliefs, political affiliation, and rural/urban residence, can influence vaccine acceptance. Including these variables could provide a more comprehensive analysis.
In conclusion, this study contributes to understanding the psychological correlates of COVID-19 vaccine acceptance. However, addressing the abovementioned issues would significantly strengthen the paper and its potential impact on public health.
Author Response

(The authors gave the same response as above.)

Round 2
Reviewer 2 Report
Thanks for the authors for addressing all reviewer's comments and recommendations.
However, I recommend to expand the methods and results section and describe it in more detail. Also I recommend to look at the paper https://doi.org/10.1038/s41586-022-05398-2 in context of information about COVID-19 accessability worldwide.
Author Response
To the Editor
Medicina July, 21th, 2023
Ref.: Manuscript ID: medicina-2505473
Title: The role of hyperarousal and aberrant salience in the acceptance of anti-COVID-19 vaccination.
Dear Editor,
Again, thank you for the opportunity to revise our above-referenced paper. The following letter gives our responses to the Reviewer’s comments. In this revised version of the manuscript, we took into consideration all comments and revised the manuscript accordingly.
Reviewers’ comments:
We wish to thank the Reviewers for reading our manuscript and reviewing it, which has helped us improve it. We next detail our responses to each reviewer’s concerns and comments.
# Reviewer 2
- “Thanks for the authors for addressing all reviewer's comments and recommendations. However, I recommend to expand the methods and results section and describe it in more detail.”
“Methods” and “Result” sections have been better detailed, as requested.
- “Also I recommend to look at the paper https://doi.org/10.1038/s41586-022-05398-2 in context of information about COVID-19 accessability worldwide.”
We thank the reviewer for suggesting the interesting paper https://doi.org/10.1038/s41586-022-05398-2 which has been added to the references.
We appreciate all your insightful comments, and we tried to be responsive to them.
Best regards.
Carmela Mento